# Scale-Up of Dark Fermentative Biohydrogen Production by Artificial Microbial Co-Cultures

İpek Ergal [1,*,†], Elisa Zech [1,†], Nikola Hanišáková [2], Ivan Kushkevych [2], Werner Fuchs [3], Tomáš Vítěz [2], Monika Vítězová [2], Günther Bochmann [3] and Simon K.-M. R. Rittmann [1,*]

[1] Archaea Physiology and Biotechnology Group, Department of Functional and Evolutionary Ecology, University of Vienna, Djerassiplatz 1, 1030 Vienna, Austria; elisa.zech@gmx.at

[2] Laboratory of Anaerobic Microorganisms, Section of Microbiology, Department of Experimental Biology, Faculty of Science, Masaryk University, Kamenice 5, 62500 Brno, Czech Republic; 446645@mail.muni.cz (N.H.); kushkevych@mail.muni.cz (I.K.); vitez@sci.muni.cz (T.V.); vitezova@sci.muni.cz (M.V.)

[3] Institute of Environmental Biotechnology, Department for Agrobiotechnology, University of Natural Resources and Life Sciences Vienna, Konrad Lorenz-Str. 20, 3430 Tulln, Austria; werner.fuchs@boku.ac.at (W.F.); guenther.bochmann@boku.ac.at (G.B.)

\* Correspondence: ipek.ergal@univie.ac.at (İ.E.); simon.rittmann@univie.ac.at (S.K.-M.R.R.)

† These authors contributed equally to this work.

**Abstract:** As a renewable energy carrier, dark fermentative biohydrogen ($H_2$) represents a promising future alternative to fossil fuels. Recently, the limited $H_2$ yield of 4 moles of $H_2$ per mole glucose, the so-called "Thauer limit", was surpassed by a defined artificial consortium. In this article, we demonstrate the upscaling of this drawing board design, from serum bottles to laboratory scale bioreactors. Our results illustrate that this designed microbial co-culture can be successfully implemented in batch mode, with maximum $H_2$ yields of 6.18 and 4.45 mol mol$^{-1}$ substrate. Furthermore, we report volumetric $H_2$ productivities of 105.6 and 80.8 mmol $H_2$ L$^{-1}$ h$^{-1}$. These rates are higher than for any other dark fermentative $H_2$ production system using a synthetic microbial co-culture applied in batch mode on a defined medium. Our study is an important step forward for the application of artificial microbial consortia in future biotechnology and energy production systems.

**Keywords:** bioreactor; bioprocess; artificial ecosystem; bacteria; mesophile

## 1. Introduction

Molecular hydrogen ($H_2$) is an energy carrier with high combustion yields [1]. Biologically produced $H_2$, referred to as biohydrogen production (BHP), is considered an environmentally friendly clean alternative with near zero carbon emission and has potential to replace fossil fuels as energy carriers [2]. Comparing different BHP systems, two main parameters must be considered. The $H_2$ evolution rate (HER/mmol $H_2$ L$^{-1}$ h$^{-1}$) represents the volumetric productivity over time and is independent of the respective culture used, as opposed to the substrate conversion efficiency ($Y_{(H2/S)}$/mol $H_2$ mol$^{-1}$ substrate). Taking these units into consideration, the high HER, rapid cell growth, and relatively simple implementation due to non-requirement of light energy, advocate the use of dark fermentative $H_2$ production (DFHP) over photobiological $H_2$ production processes [3,4]. However, the low $Y_{(H2/S)}$ is the major drawback of DFHP, which is restricted to a theoretical maximum of 4 moles $H_2$ produced per one mole of glucose consumed in microbial pure cultures and microbial enrichment cultures when acetate is produced as a by-product [5]. DFHP can be carried out by various organisms using mainly two different $H_2$ generating pathways. Strictly anaerobic $H_2$ producers perform the pyruvate ferredoxin oxidoreductase (PFOR) pathway, whereas the pyruvate formate lyase (PFL) pathway is active in facultative anaerobes [3]. As the name already implies, in the course of the PFOR pathway, $H_2$ is generated by a ferredoxin dependent hydrogenase enzyme. Depending on the organism, reduction equivalents may originate from glycolysis or from the conversion of pyruvate to

acetyl-CoA and reduced ferredoxin [6]. Alternatively, pyruvate is converted into formate via the activity of the PFL enzyme, and formate is then split into $H_2$ and $CO_2$ [6].

These two main $H_2$ generating metabolic routes are active in multiple microbial species. Members of Enterobacteriaceae and Clostridiaceae are both very extensively studied and successful $H_2$ producing microbes [3,7,8]. Among these, *Enterobacter aerogenes* and *Clostridium acetobutylicum* have shown high $H_2$ productivities in pure culture. Nevertheless, the maximum $Y_{(H2/S)}$ of 3.14 and 2.16 mol $H_2$ mol$^{-1}$ substrate, by *C. acetobutylicum* and *E. aerogenes*, respectively [9,10], are still below the theoretical limit.

Interspecies interactions within microbial communities have shown positive effects on the productivity of fermentation systems [11,12]. Hence, *E. aerogenes* and *C. acetobutylicum* were grown in a co-culture as an attempt to increase the $Y_{(H2/S)}$. This defined artificial microbial consortium surmounted the restriction of 4 moles $H_2$ mol$^{-1}$ glucose in DFHP [13]. In a drawing board like approach to establish a pipeline for design and engineering of artificial microbial consortia for DFHP [14], the cultivation parameters were pre-selected by considering a priori physiological and biotechnological knowledge from a preceding meta-data analysis [3]. With the design of experiments (DoE) approach, a mutual medium was developed, taking into consideration each of the organisms' nutritional requirements and the buffer capacity of the medium. In addition, refinement of initial substrate and cell concentrations were performed to prevent substrate inhibition and ensure a stable coexistence of the two organisms. This precision design of an artificial microbial consortium has resulted in the proliferation of both organisms and in exceeding the previously described physiological limit by reaching a $Y_{(H2/S)}$ of 5.58 mol mol$^{-1}$ [13]. As with most studies on BHP, this study was also conducted in serum bottles, with a closed batch cultivation mode without controlling cultivation parameters, such as pH. Being fast and simple in application, the cultivation in closed systems can be used for screening of the microbial strains used [4]. Yet, DFHP in closed batch systems are limited in their growth and $H_2$ productivity due to a decrease in pH and high $H_2$ partial pressure, which reduces HER [15]. To overcome these drawbacks, $N_2$ sparging and pH control [16,17] may be applied to enhance DFHP [18]. Thus, the DFHP should be performed in bioreactors to assess their suitability for subsequent scale-up.

Even though DFHP has already been investigated for more than a century [3,4,19], there are only a limited number of results of batch and in continuous culture experiments available for pure cultures and for defined microbial DFHP ecosystems. While the bioprocess parameters e.g., pH, substrate concentration, and temperature can be controlled, the identification of suitable scale-up procedures and parameters in batch cultivation mode, as well as data on long-term bioprocess stability of BHP in continuous culture, are urgently required. Ideally, after screening for the basic requirements during closed batch cultivation, the microbes can then be implemented in laboratory scale bioreactors to examine their physiological potential for high quantitative BHP. These insights will help for future scale-up of the process to pilot scale bioreactors and possible industrialization. It has already been shown that cultivation in batch mode can increase $H_2$ productivity compared to closed batch cultivation [18] and that careful strain selection and optimization of the culture conditions genuinely affect the bioreactor performance [20]. Besides, some successful pilot scale DFHP experiments have been performed already. $Y_{(H2/S)}$ of 2.12 and 2.76 mol $H_2$ mol$^{-1}$ glucose were obtained by a consortium of *C. butyricum* and *C. pasteurianum* in 20 L batch bioreactors [7] and a tri-culture of *Citrobacter freundii*, *Enterobacter aerogenes*, and *Rhodopseudomonas palustris* in a 100 L vessel [21], respectively. Two consortia of *Enterobacter cloacae* plus *Bacillus cereus* and *E. cloacae* plus *Klebsiella* sp., produced 3.2 and 3 mol $H_2$ per mol glucose, respectively, at a working volume of 4 L in 5 L bioreactors [22].

The aim of this study was to examine if a drawing board like design of an artificial microbial DFHP co-culture can be propagated towards future industrial scale fermentation processes. Therefore, the scale-up experiment was carried out in laboratory-scale bioreactors rather than closed batch serum bottles, to follow a gradual scale up strategy. Our approach differs from the previously applied DFHP experiments regarding a re-assessment

of the inoculation ratio of the two microorganisms, described in our most recent paper [14]. Furthermore, apart from examining the performance in volumetrically larger vessels, our experimental set-up enabled the manual and controlled adjustment of the most crucial fermentation variables, including pH, temperature, $N_2$ gassing rate, and agitation speed. We hypothesized that a defined microbial consortium of *E. aerogenes* and *C. acetobutylicum* can be scaled-up regarding HER/mmol $H_2$ $L^{-1}$ $h^{-1}$ and $Y_{(H2/S)}$/mol $mol^{-1}$, from closed batch to batch.

## 2. Materials and Methods

### 2.1. Chemicals

$CO_2$, $N_2$, and $H_2$ were 99.999 Vol.$-$%. In addition, 20 Vol.$-$% $CO_2$ in $N_2$ was used (Air Liquide, Schwechat, Austria). All other chemicals were of highest grade available.

### 2.2. Experimental Set-Up

Cultures of *Clostridium acetobutylicum* DSM 792 and *Enterobacter aerogenes* DSM 30053 were used for pure culture and consortium experiments. Both microorganisms were cultivated strict anaerobically in a DASGIP parallel bioreactor system in 2 L bioreactors (Eppendorf AG, Hamburg, Germany) using 1.5 L working volume over the time period from 40 to 53 h until the $H_2$ production ceased. A defined medium was used for all the experiments (including pre-cultures), as previously described in detail elsewhere [13], containing (per L) 3.47 g of $NH_4Cl$, 10.41 g of $KH_2PO_4$, 5.31 g of $K_2HPO_4$, and 1.35 g of NaCl. To each bioreactor, 7.5 mL of a 200× vitamin stock solution was added, containing (per L) 0.2 g of 4-amino-benzoic acid, 0.9 g thiamine, 0.002 g biotin, as well as 15 mL of a 100× mineral stock solution containing (per L) 0.2 g of $MgSO_4\cdot7H_2O$, 0.01 g of $MnCl_2\cdot4H_2O$, 0.01 g of $FeSO_4\cdot7H_2O$, and 0.01 g of NaCl. Glucose served as carbon source for $H_2$ production batch experiments at a concentration of 30 g $L^{-1}$. Before inoculation, glucose and mineral solution were sterilized separately at 121 °C for 20 min, and vitamin solution was sterilized by filtration (0.2 µm pore size). Anaerobic conditions inside the bioreactors were obtained by flushing the vessels with $N_2$ prior to inoculation. Pre-cultures were grown anaerobically at 0.3 bar in a $N_2$ atmosphere in a closed batch set-up. Inoculation was performed using the required amount of *C. acetobutylicum* pre-culture to reach an optical density of 0.3 in the bioreactor (ranging from 150 to 200 mL) and 0.01% (*v/v*) of *E. aerogenes* DSM 30053 (15 mL) of an anaerobically and aseptically transferred inoculum from the pre-culture vessels to the bioreactor.

The experiments were performed once (N = 1) with controlled pH and twice (N = 2) with uncontrolled pH, and both sets were performed in duplicates (*n* = 2). Temperature was set at 37 ± 0.5 °C, agitation speed at 100 and 200 rpm, and $N_2$ inflow rate at 1 sL $h^{-1}$. A pH probe (Mettler Toledo GmbH, Wien, Austria) and a redox probe (Mettler Toledo GmbH, Wien, Austria) were used to observe the pH and oxidation reduction potential (ORP), respectively.

### 2.3. Optical Density (OD) Measurements and Cell Counting

At each time point, 1 mL of liquid sample was collected from the bioreactors and the optical density (600 nm (OD600)) was measured with a spectrophotometer (Specord 200 Plus, AnalytikJena, Jena, Germany). After increased growth of the culture, the samples were diluted 1:10 to ensure an exact measurement in the linear absorption range.

To determine the cell concentrations in the pre-cultures, 1 mL samples were retrieved using sterile syringes (Soft-Ject, Henke SassWolf, Tuttlingen, Germany) and hypodermic needles (Sterican size 14, B. Braun, Melsungen, Germany).

Cells were counted using a Nikon Eclipse 50i microscope (Nikon, Amsterdam, Netherlands) at each sampling point. 12 µL of each sample (non-, 1:10, 1:50 or 1:100 diluted) were applied onto a Neubauer improved cell counting chamber (Superior Marienfeld, Lauda-Königshofen, Germany) with a grid depth of 0.1 mm.

### 2.4. Quantification of Gas Composition

Gas samples were taken in a gas bag (10 L SKC Quality Sampling Bag, SKC Inc., Covington, GA, USA) connected to the off-gas tubing at each time point. Once the gas bag was filled, the gas was collected and transferred into sealed (Butyl rubber 20 mm, Chemglass Life Science LLC, Vineland, NJ, USA) and crimped 120 mL glass serum bottles (Ochs Glasgerätebau, Langerwehe, Germany) which were flushed with the fermentation off-gas for 5 min applying hyperdermic needles (Sterican size 14, B. Braun, Melsungen, Germany) and appropriate tubing.

The compositions of the collected gas samples were analyzed using gas chromatography (GC) (7890A GC System, Agilent Technologies, Santa Clara, CA, USA) with a 19,808 Shin Carbon ST Micropacked Column (Restek GmbH, Bad Homburg, Germany). The measurements were accomplished with a gas injection and control unit (Joint Analytical System GmbH, Moers, Germany), as described before [23]. A thermal conductivity detector was used for the measurements, and the gases were separated at 170 °C using helium as the carrier gas. The reference flow setting was 10 mL min$^{-1}$. The makeup flow was set to 1 mL min$^{-1}$. The standard gasses for GC measurements were 99.999 Vol.−% $H_2$, 99.999 Vol.−% $CO_2$, 99.999 Vol.−% $N_2$, 20 Vol.−% $CO_2$ in $H_2$, 20 Vol.−% $CO_2$ in $N_2$, a test gas containing 4.5 Vol.−% $H_2$ in $N_2$, and a test gas containing 22.4 Vol.−% $H_2$; 19.7 Vol.−% $CO_2$; 15.5 Vol.−% $N_2$ 14.1 Vol.−% $CH_4$ in CO, and a test gas containing 22.4 Vol.−% $H_2$; 19.7 Vol.−% $CO_2$; and 12.2 Vol.−% $N_2$ (Air Liquide GmbH, Schwechat, Austria). Another standard test gas for GC measurements comprised the following composition: 0.01 Vol.−% $CH_4$; 0.08 Vol.−% $CO_2$ in $N_2$ (Messer GmbH, Wien, Austria). Standard GC curves with an $R^2$ of 0.99 or higher were obtained with aforementioned standard gases.

### 2.5. Quantification of Metabolites

The concentrations of sugars, volatile fatty acids, and alcohols were quantified using high-performance liquid chromatography (HPLC). Measurements were done at 45 °C using an Agilent 1100 system consisting of a G1310A isocratic pump, a G1313A ALS autosampler, a Transgenomic ICSep ICE-ION-300 column, a G1316A column thermostat set at 45 °C, and a G1362A RID refractive index detector. Then, 40 μL of sample was used as injection volume and 0.005 mol L$^{-1}$ $H_2SO_4$ as solvent, with a flow rate of 0.325 mL min$^{-1}$ and a pressure of 48–49 bar.

### 2.6. DNA Extraction and qPCR

DNA was extracted from 1 mL culture samples at each time point as follows: After centrifugation (at 4 °C and 13,400× *g* revolutions per minute (rpm) for 30 min) and resuspension in pre-warmed (65 °C) 1% sodium dodecyl sulfate (SDS) extraction buffer, the cells were transferred to Lysing Matrix E tubes (MP Biomedicals, Santa Ana, CA, USA) containing equal volume of phenol/chloroform/isoamyl alcohol (25:24:1) and around 0.5 g Bulk B Beads, and lysed in a FastPrep-24 (MP-Biomedicals, NY, USA) device (speed setting 4 for 30 s). This was followed by centrifugation at 13,400× *g* rpm for 10 min. An equal volume of chloroform/isoamyl alcohol (24:1) was added to the supernatant, and the mixture was then centrifuged again at 13,400× *g* rpm for 10 min. Addition of 1 μL glycogen (20 mg mL$^{-1}$) and double volume of polyethylene glycol (PEG) solution (30% PEG, 1.6 mol L$^{-1}$ NaCl) allowed the DNA to precipitate, which was performed overnight at 4 °C. Nucleic acid pellets were retrieved by centrifugation at 13,400× *g* rpm for 30 min, followed by washing with 70% cold ethanol solution, drying in a SpeedVac at 30 °C (Thermo Scientific, Dreieich, Germany), and resuspension in 40 μL Tris-EDTA buffer. The extracted DNA was stored at −20 °C until further analysis. Quantification of Nucleic Acid was performed with NanoDrop ND-1000 spectrophotometer (NanoDrop Technologies, Wilmington, DE, USA).

For the qPCR diluted DNA equivalents (1:300) were used for analysis. Additionally, negative controls with sterile DEPC water as a replacement for the DNA templates were run in parallel. Six standards with previously determined cell concentrations at different dilutions, ranging from 1:10 to 1:1·10$^6$, were amplified simultaneously and used both

as a positive control and to generate a standard curve as described elsewhere [13]. All amplification reactions were run in triplicates.

To prevent false positive amplification, primer design was done by targeting species specific genes. Using the ClustalW2 multiple sequence alignment program (http://www.ebi.ac.uk/Tools/clustalw2/, accessed on 15 March 2018), optimal primers were identified by sequence comparison of the genes.

For *E. aerogenes* DSM 30053 forward primer 5′GCG TTG TGG GGT TGC ACG AT 3′ and reverse primer 5′ TGG CGC GCG AGC ACA TTT TC 3′; for *C. acetobutylicum* DSM 792 forward primer 5′ TGG CAC AGT CAG TCG GCT ACC 3′; and reverse primer 5′ GCG TGA TGC ACC TAA CCC AGC 3′ were used.

Reactions were set up using SYBR Green labelled Luna Universal qPCR Master Mix (M3003L, New England Biolabs, Ipswich, MA, USA) following the manufacturer's protocol and performed in Eppendorf Mastercycler epgradientS realplex2 (Eppendorf, Hamburg, Germany). Amplification protocol was run, as described in detail elsewhere [13].

### 2.7. Data Analysis

To determine the specific growth rate ($\mu/h^{-1}$) for each bioreactor, the following equation was used: $X = X^0 \cdot e^{\mu t}$ with X, cell number/cells $mL^{-1}$; $X^0$, initial cell number/cells $mL^{-1}$; t, time/h; and e, Euler number. Calculation of HER/mmol $H_2$ $L^{-1}$ $h^{-1}$ was done by taking into consideration the total gas flow rate/sL $h^{-1}$, the respective concentration of $H_2$, the ideal gas law, and the inert gas flow correction factor.

## 3. Results

### 3.1. Experimental Set-Up

To establish the artificial consortium of *C. acetobutylicum* and *E. aerogenes* in bioreactors, the culturing conditions were at large kept identical to those described before [13]. We anticipated that the initial ratio that was optimized for the closed batch runs had to be adjusted to the batch set-up. Nevertheless, neither an increase nor a decrease of the initial *E. aerogenes* cell concentration (ratios ranging from 1:10 to 1:10,000,000; *E. aerogenes*: *C. acetobutylicum*) resulted in stable growth or significant $H_2$ production (Figure 1). This initial inoculation ratio of 1:10,000 in favour of *C. acetobutylicum* can thus be considered the optimum inoculation ratio both for closed batch and batch cultivation. Therefore, medium, substrate, and cell concentrations were kept the same for the up-scaling experiments in the bioreactors. To further optimise $H_2$ production, we adjusted the system pH. During the first cultivations, the initial pH was set to 6. However, a rapid pH decrease was observed due to the accumulation of acidic metabolic end products for the experiments with uncontrolled pH. When pH was controlled at 6, an increase in $Y_{(H2/S)}$ and HER were observed (Figure 2).

### 3.2. $H_2$ Production

The highest $Y_{(H2/S)}$ of 6.18 mol $H_2$ $mol^{-1}$ substrate and HER of 105.6 mmol $H_2$ $L^{-1}$ $h^{-1}$ were achieved when the pH was controlled in the interval between 20–25 h after inoculation. Whereas the second-highest $Y_{(H2/S)}$ of 4.45 mol $H_2$ $mol^{-1}$ substrate and HER of 80.8 mmol $H_2$ $L^{-1}$ $h^{-1}$ were observed under non-controlled conditions between 25 to 29 h after inoculation (Figure 2). Both values clearly surpass the theoretical limit of 4 mol $H_2$ $mol^{-1}$ glucose and also the highest $Y_{(H2/S)}$ and HER values that had been obtained in closed batch before. We have to stress here that these high values have only been obtained within the respective intervals between two timepoints, taking into consideration not only the glucose consumption but also the re-uptake of metabolic by-products (Supplementary Materials Table S1).

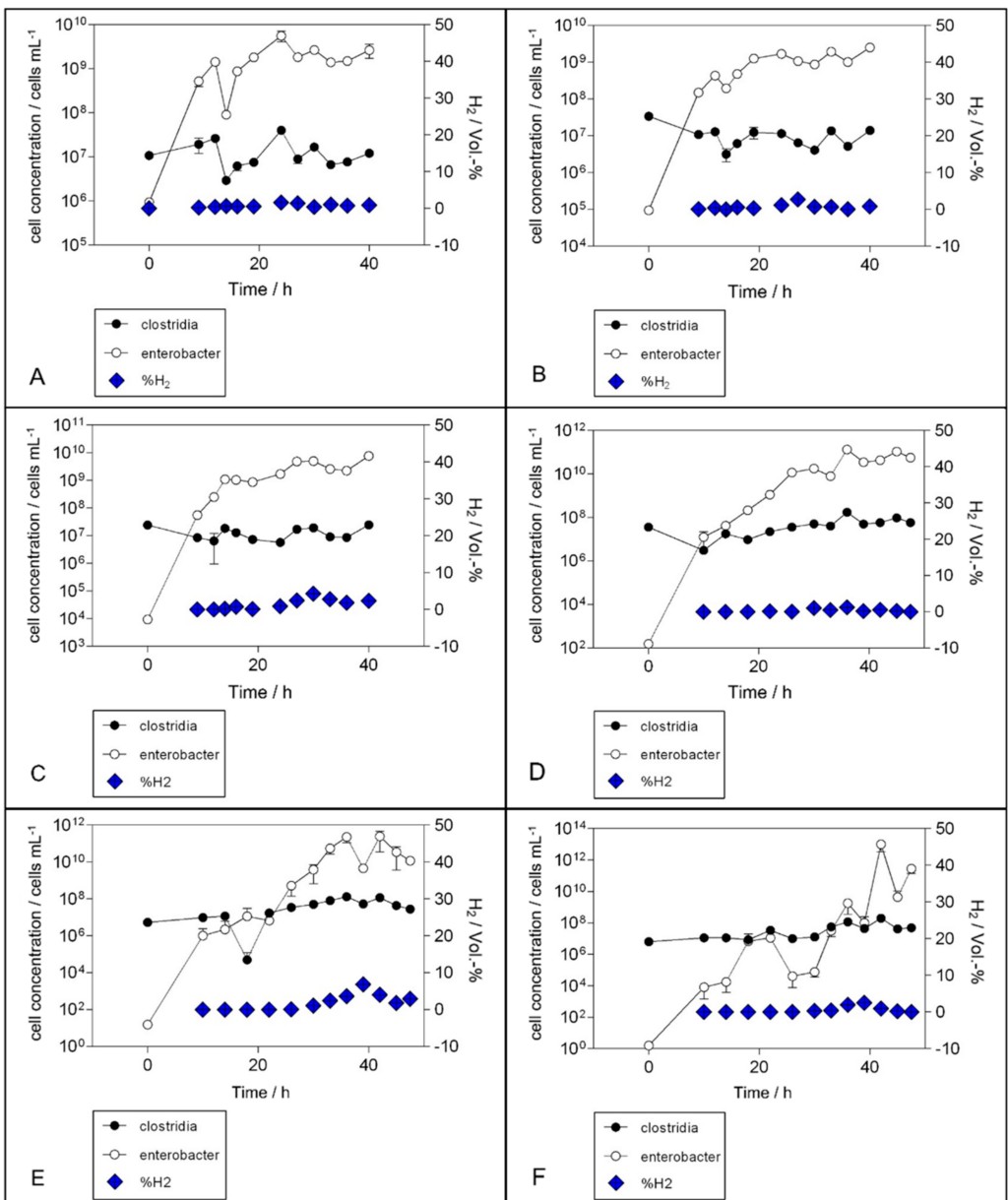

**Figure 1.** Different bioreactor runs with varying initial cell concentrations. (**A**) 1:10; (**B**) 1:100; (**C**) 1:1000; (**D**) 1:100,000; (**E**) 1:1,000,000; and (**F**) 1:10,000,000 for *C. acetobutylicum*: *E. aerogenes*. Depicted are the respective cell concentrations of *Clostridium acetobutylicum* and *Enterobacter aerogenes*, measured with qPCR, as well as the $H_2$ concentration.

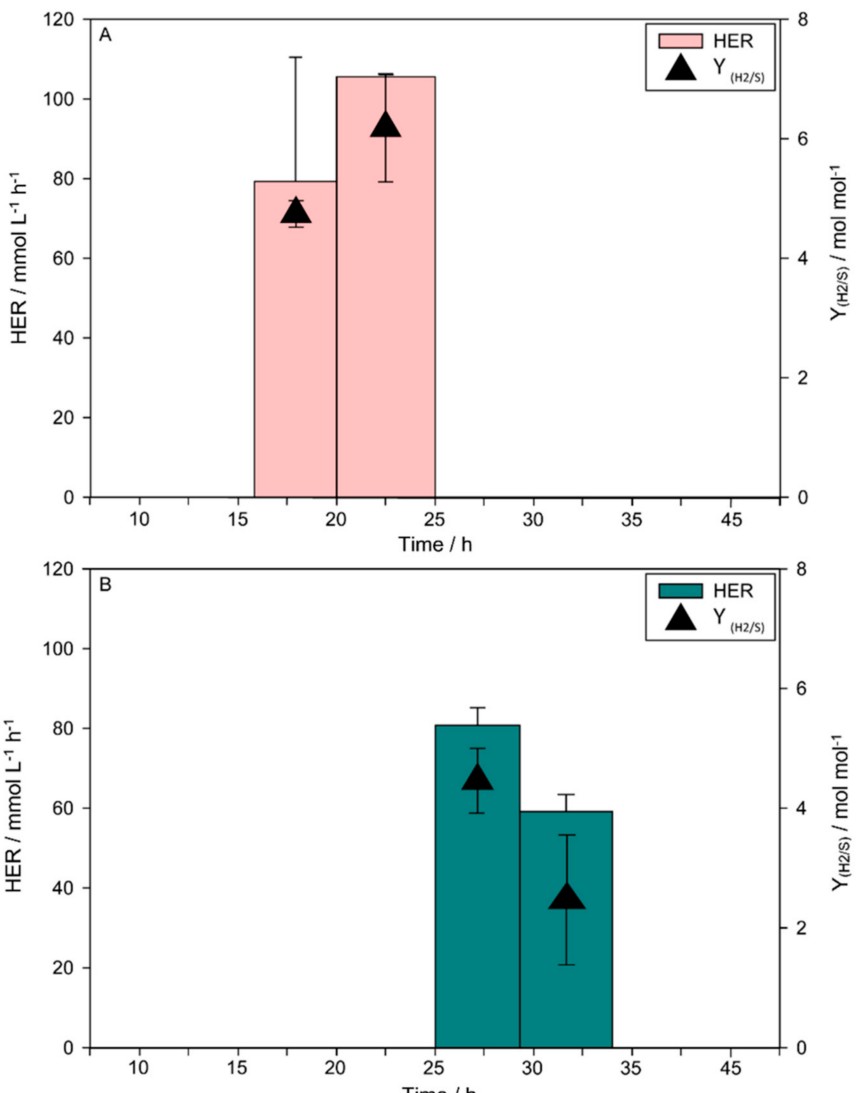

**Figure 2.** HER and $Y_{(H2/S)}$ results over time for the two best bioreactor runs. The displayed results were achieved when the pH was kept stable at 6 in the top chart (**A**), and when the pH was not controlled in the lower chart (**B**). Bars indicate HER; single data plots represent the $Y_{(H2/S)}$; standard deviations are given as error bars.

## 4. Discussion

To our knowledge, this is the first report of a synthetic co-culture cultivated in batch mode describing an improvement of $Y_{(H2/S)}$ beyond the Thauer limit. For comparison, Table 1 lists different studies on DFHP using synthetic consortia operated in batch cultivation systems.

With the application of *Caldicellulosiruptor kristjanssonii* and *Caldicellulosiruptor saccharolyticus* Zeidan and Van Niel [24] reported a $Y_{(H2/S)}$ that is very close to the theoretical limit (3.8 mol $H_2$ $mol^{-1}$ C6 sugar equivalent). This result was achieved by using extreme thermophilic organisms which, due to thermodynamics, usually produce higher yields than mesophiles [7,25]. Furthermore, and like all other studies on co-cultures listed in Table 1, the experiments were conducted using a complex medium rather than a defined medium, making a comprehensive $H_2$ production analysis challenging.

Apart from using complex or defined media, earlier reports on DFHP using artificial consortia in batch reactors deviate from the current study in the working volume within the

bioreactor. Working volumes range from only 100 mL [9] to as much as 18 L [7] (Table 1). Also varying among the different studies is the pH, ranging from 5.25 [26] to 7 [9,27].

Optimum pH values for maximum $H_2$ production were found to be slightly acidic, around 6.5 [24,28,29]. This parameter directly affects the hydrogenase activity, metabolic by-products, and $Y_{(H2/S)}$ [30], which is why the ability to monitor and control the in situ acidity/alkalinity during the fermentation procedure is a very convenient feature of the batch fermentation set-up.

*Clostridium* sp. have been shown to stably produce $H_2$ at a pH of 5.5–6 [31,32], while the peak substrate conversion rate of *E. aerogenes* was found between pH 6 and 7 [33]. With no base or acid inflow to keep the pH steady, we observed a rapid acidification of the medium due to acidogenic growth properties of the organisms. This acidification lead, at first, to an increased $H_2$ production, but very acidic conditions will eventually initiate a metabolic switch to solventogenesis, or even the formation of spores in *Clostridium* sp. with decreased $H_2$ productivities. Both spore formation and production of solvents rather than acids can be seen as preventive actions to keep harmful effects of undissociated acids at bay [34,35]. Hence, keeping the pH stable at 6 would favour the acetate pathway and prevent the metabolic shift. Our results confirm this assumption, as the maximum $Y_{(H2/S)}$ (6.18 mol $H_2$ mol$^{-1}$ substrate) was observed under controlled pH conditions, and a lower $Y_{(H2/S)}$ (4.45 mol $H_2$ mol$^{-1}$ substrate) was achieved when the pH was allowed to decrease. Interestingly, this second-best result was obtained rather late in the experiment (Figure 2) when the pH dropped below the optimum value.

In addition to the pH, other environmental parameters have a crucial influence on the system performance. Ergal et al. [13] found that only a remarkable discrepancy in the initial cell concentration allowed the stable co-existence of *C. acetobutylicum* and *E. aerogenes*. This was necessary, as the fast-growing *E. aerogenes* threatened to quickly overgrow the *C. acetobutylicum* population. Since different strains show different growth behaviour, it is important to counteract possible imbalances by compensating with varying initial cell concentrations. Once the proliferation of subdominant species can be guaranteed, the requirements for a stable and well-functioning synthetic consortium were provided. Yet, the initial cell ratio of 1:10,000 that enabled high $H_2$ yield and productivity in Ergal et al. [13] is quite unique. Usually, inoculation ratios do not exceed 1:1 or 1:2 [7,22,29]. Within this work, we show that the designed co-culture can be successfully applied and produced $Y_{(H2/S)}$ beyond theoretical limits after the initial report in closed batch mode in serum bottles, and also performed here in batch mode in bioreactors.

Synthetic microbial consortia are, in fact, well applicable for increased $H_2$ production, compared to monocultures. Still, the theoretical limit of 4 mol $H_2$ per mol glucose during DFHP can barely be met and is almost never exceeded. In this regard, thermophilic strains are more favourable over mesophilic ones, as higher temperatures favour increased specific $H_2$ productivities [24,36]. Mesophilic $H_2$ producers have the advantage of being extensively studied, and the literature provides a broad and detailed understanding of their physiological and genetic properties. These insights are required to design specific microbial consortia and set up a suitable environment that meets the organisms' requirements and enables high biofuel production. Following this drawing board like approach, Ergal et al. [13] succeeded to implement successful synthetic microbial consortia producing $Y_{(H2/S)}$ that surpassed the Thauer limit. It was also what distinguished their study from other reports on BHP by artificial consortia.

**Table 1.** Overview of studies on DFHP with pure cultures and co-cultures.

| Microorganism | Feeding Substrate | pH | Temperature/$^\circ$C | Medium Composition (Complex/Defined) | $Y_{(H2/S)}$/mol mol$^{-1}$ | HER/mmol L$^{-1}$ h$^{-1}$ | Operating Conditions | Ref. |
|---|---|---|---|---|---|---|---|---|
| Pure cultures | | | | | | | | |
| *Clostridium acetobutylicum* | glucose | 7 | 30 | complex | 3.14 | NA | 100 mL in 500 mL Schott bottle | [9] |
| *Clostridium acetobutylicum* | cassava waste-water | 7 | 36 | complex | 2.41 | NA | 300 mL bioreactor | [27] |
| *Clostridium acetobutylicum* | sugarcane molasses | 6.5 | 30 | complex | 1.3 | NA | 1950 mL in 2 L MultiGen fermenter | [37] |
| *Enterobacter aerogenes* | maltose | 6.5 | 35 | complex | 2.16 | NA | 52 mL culture in Erlenmeyer flask | [10] |
| *Enterobacter aerogenes* | corn starch | 5.5 | 40 | complex | 1.8 | 5.2 | 1.5 L in 2 L Gallenkamp FBL-195 bioreactor | [38] |
| *Enterobacter aerogenes* | glucose | uncontrolled (initial pH 6.9) | 37 | defined | 1.36 | NA | 3 L in 5 L bioreactor | [39] |
| *Enterobacter aerogenes* | glucose | 6.8 | 30 | defined | 0.55 | 39.92 | 0.8 L in 2 L table-top bioreactor | [40] |
| *Enterobacter aerogenes* | corn starch | 6.5 | 39 | complex | 1.09 | 17.39 | 1.5 L in 2 L Gallenkamp FBL-195 bioreactor | [33] |
| Co-cultures | | | | | | | | |
| *Clostridium acetobutylicum* and *Desulfovibrio vulgaris* | glucose | NA | 37 | complex | 3.46 | NA | cultivated in Hungate tubes | [24] |

**Table 1.** *Cont.*

| Microorganism | Feeding Substrate | pH | Temperature/°C | Medium Composition (Complex/Defined) | $Y_{(H2/S)}$/mol mol$^{-1}$ | HER/mmol L$^{-1}$ h$^{-1}$ | Operating Conditions | Ref. |
|---|---|---|---|---|---|---|---|---|
| *Clostridium butyricum* and *Enterobacter aerogenes* | sweet potato starch | 5.25 | 37 | complex | 2.4 | NA | 200 mL in 250 mL stirred reactor | [26] |
| *Clostridium butyricum* and *Clostridium pasteurianum* | starch | 5.3 | 30 | complex | 2.32 | NA | 18 L in 20 L stainless steel tank bioreactor | [7] |
| *Clostridium butyricum* and *Clostridium pasteurianum* | glucose | 5.3 | 30 | complex | 2.12 | NA | 18 L in 20 L stainless steel tank bioreactor | [7] |
| *Klebsiella pneumoniae* and *Citrobacter freundii* | glucose | 6.5 | 37 | complex | 2.07 | NA | 2 L in a controlled fermenter | [29] |
| *Enterobacter aerogenes* and *Clostridium acetobutylicum* | glucose | 6 | 37 | defined | 6.18 | 105.59 | 1.5 L in 2 L stirred tank reactor | This study |
| *Enterobacter aerogenes* and *Clostridium acetobutylicum* | glucose | uncontrolled | 37 | defined | 4.45 | 80.76 | 1.5 L in 2 L stirred tank reactor | This study |

## 5. Conclusions

Artificial microbial ecosystems can be effectively used for scale-up of DFHP from closed batch to lab scale bioreactors. In this study, we obtained the highest $Y_{(H2/S)}$ and the highest HER on glucose for any DFHP system using a synthetic microbial co-culture on a defined medium in batch mode up to date. This work provides the fundamentals for further scale-up of our DFHP bioprocess, which is required to unravel the scaling criterion aiming to establish DFHP at industrial scale. Further studies on design and engineering of artificial microbial consortia for DFHP regarding substrates such as lignocellulose, lipid waste, and food waste will drive our understanding of their function. Moreover, the development of more sophisticated techniques to control the physical space and environment of engineered microbial consortia might lead to a further improvement of HER and $Y_{(H2/S)}$. This study is another step forward in demonstrating the application possibilities of artificial microbial consortia in future biotechnology and energy production systems.

**Supplementary Materials:** The following supporting information is available online at https://www.mdpi.com/article/10.3390/applmicrobiol2010015/s1. Table S1: Dataset.

**Author Contributions:** Conceptualization, İ.E., W.F., T.V., M.V., G.B. and S.K.-M.R.R.; methodology, I.K., W.F., M.V., G.B. and S.K.-M.R.R.; validation, İ.E. and S.K.-M.R.R.; formal analysis, İ.E.; investigation, İ.E., E.Z., N.H. and I.K.; resources, W.F., M.V., G.B. and S.K.-M.R.R.; data curation, İ.E. and E.Z.; writing original draft preparation, İ.E., E.Z. and S.K.-M.R.R.; writing review and editing, İ.E., E.Z., I.K., W.F., T.V., M.V. and S.K.-M.R.R.; visualization, İ.E. and E.Z.; supervision, W.F., M.V., G.B. and S.K.-M.R.R.; project administration, M.V. and S.K.-M.R.R.; funding acquisition, W.F., T.V., M.V., G.B. and S.K.-M.R.R. All authors have read and agreed to the published version of the manuscript.

**Funding:** This research was funded by the Austrian Research Promotion Agency (Forschungsförderungsgesellschaft (FFG)), project H2.AT, grant number 853618; and funded by BMBWF, project WTZ, grant number CZ 08/2020.

**Institutional Review Board Statement:** Not applicable.

**Informed Consent Statement:** Not applicable.

**Data Availability Statement:** The datasets used and/or analysed during the current study are available from the corresponding author on reasonable request.

**Conflicts of Interest:** The authors declare no conflict of interest.

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
