# Peer review of "Scale-Up of Dark Fermentative Biohydrogen Production by Artificial Microbial Co-Cultures"

_2673-8007, doi:10.3390/applmicrobiol2010015_

Round 1

Reviewer 1 Report

The authors performed the scale-up of dark fermentative biohydrogen production using two-difference microbes, Clostridium acetobutylicum and Enterobacter aerogenes. Specific comments are as follow. 

1. Introduction  
Please complement research novelty compared to previous studies. 

2. I recommend that divide Results and discussion section with appropriate sub-chapter (ex> 3.1. 3.2. 3.3...). It would be helpful to understand.

3. Check typo errors throughout the manuscript. 

4. Figure S1 seems to be more appropriate as main figure (not supplementary figure). Revise Figure S1 and complement discussion in the manuscript. 

Author Response

Reviewer 1

Comments and Suggestions for Authors

The authors performed the scale-up of dark fermentative biohydrogen production using two-difference microbes, Clostridium acetobutylicum and Enterobacter aerogenes. Specific comments are as follow.

1. Introduction
Please complement research novelty compared to previous studies.

Thank you for your comment. The novelty is now provided at the end of the introduction together with the goals and hypothesis.

2. I recommend that divide Results and discussion section with appropriate sub-chapter (ex> 3.1. 3.2. 3.3...). It would be helpful to understand.

Thank you for your comment. We changed the sections in the manuscript accordingly.

L227: “3. Results”

L228: “3.1. Experimental set-up”

L246: “3.2. H2 production”

L263: “4. Discussion”

3. Check typo errors throughout the manuscript.

Thank you for your comment. Any typo errors or misspellings have been corrected throughout the manuscript.

4. Figure S1 seems to be more appropriate as main figure (not supplementary figure). Revise Figure S1 and complement discussion in the manuscript.

Thank you for your comment. We revised Figure S1 as a main figure and adjusted the text accordingly.

Reviewer 2 Report

I reviewed article entitled “Scale-up of dark fermentative biohydrogen production by artificial microbial co-cultures” by   Ergal et al.  Although I am neophyte of the subject matter I enjoyed reading and get to know material presented in the manuscript. This is an experimental study  which was well carried out. Objectives and  findings of the work were discussed seamlessly. I am sure some of the readers of this journal will be interested in data presented in the paper. I have two minor comments or maybe typos, once they are corrected, I guess article will be ready for publishing.

LINE 138   2.3.  OD should be first spelled then (OD),  some readers may not be familiar to OD.

Figure 1:  Axes of the figure should also be fully spelled first to give a better understanding to the readers. I suggest using such format :  Time (t)-h     and others.

Author Response

Reviewer 2

Comments and Suggestions for Authors

I reviewed article entitled “Scale-up of dark fermentative biohydrogen production by artificial microbial co-cultures” by Ergal et al. Although I am neophyte of the subject matter I enjoyed reading and get to know material presented in the manuscript. This is an experimental study which was well carried out. Objectives and findings of the work were discussed seamlessly. I am sure some of the readers of this journal will be interested in data presented in the paper. I have two minor comments or maybe typos, once they are corrected, I guess article will be ready for publishing.

LINE 138 2.3. OD should be first spelled then (OD), some readers may not be familiar to OD.

Thank you for your comment. We changed L142 accordingly: “2.3. Optical density (OD) measurements and cell counting”.

Figure 1: Axes of the figure should also be fully spelled first to give a better understanding to the readers. I suggest using such format: Time (t)-h and others.

Thank you for your comment. The axes titles have been changed to “Time [h]”